# The Anuvrat Movement: A Case Study of Jain-inspired Ethical and Eco-conscious Living

**Michael Reading** [1,2]

1   Religious Studies Department, Mt. St. Mary's University, Los Angeles, CA 90049, USA;
    michael.reading@cst.edu
2   Department of Religion (Comparative Theology and Philosophy), Claremont School of Theology, Claremont,
    CA 91711, USA

**Abstract:** From proclaiming the equality of all life forms to the stringent emphasis placed upon nonviolent behavior (*ahimsa*), and once more to the pronounced intention for limiting one's possessions (*aparigraha*), Jainism has often been pointed to for its admirably ecofriendly example. Incorporating some of this eco-friendliness into its design for ethical vow taking, the Jain-inspired Anuvrat Movement, founded in 1949 by Acharya Sri Tulsi, today offers some arguably vital relevance for the urgent modern task to live eco-consciously. While such relevance includes, most explicitly, Anuvrat's final vow (vow eleven) which calls for practitioners to "refrain from such acts as are likely to cause pollution and harm the environment," and to avoid the "cutting down of trees" and the "wasting of water,"[1] it also includes several of Anuvrat's other vows as well, which carry significance on a more implicit level. Hence, presenting some of the basic history and philosophy behind Anuvrat, this article also analyzes its potential for ensuring ethical (and eco-conscious) behavior via its hallmark mechanism of vow restriction—a modality of arguably potent strategic and motivational value. Altogether, while first providing a brief inventory of Jain ecological practice in general, the article will then turn its attention to Anuvrat, arguing that when it comes to the modern eco-conscious imperative to "live simply so that others may simply live" (as the popular adage has it), there is indeed much that Anuvrat has to offer.

**Keywords:** Jainism; Anuvrat Movement; eco-conscious living; ecology; ecological vow-taking

---

Jainism has often been celebrated for its admirably ecofriendly example. As L.M. Singhvi has shown in his brief but compelling "The Jain Declaration on Nature," for instance, the various ecological aspects of this tradition are indeed numerous and widely-varying, and go well beyond *ahimsa* (nonharm to all living things) and *aparigraha* (limited possession)—the ethical and religious principles with which, in addition to *anekantavada* (non-absolutism), Jains have typically been most closely associated.[2] With this being said, at a time when the world urgently needs a greater embrace of eco-conscious simplicity as well as a far broader awakening of animal rights, the ecological value of *ahimsa* and *aparigraha* should by no means be understated. Seeing how Jains (and especially Jain mendicants) regularly take these two virtues to their utmost furthest extreme, they can naturally provide some vital insight in terms of what it looks like to dedicate oneself to, as well as meticulously execute, an eco-conscious and bio-friendly lifestyle. Still, as Singhvi and others have shown, a broader survey of Jain religious practice reveals a host of other meaningfully eco-conscious elements as well. Among others, these

---

1   (Anuvrat Global Organisation (Anuvibha) n.d.): accessible via http://www.anuvibha.in/Code-of-Conduct.htm.
2   (Singhvi 2002, pp. 217–24).

include: *jiva-daya* (compassion and charity shown to the non-human living world), a recognition of the basic interdependence of life (all organisms being in some way mutually dependent upon each other), the practice of *gupti* (self-restraint) and *samyaktva* (equanimity)—both of which, like *aparigraha*, guard against the human eco-destructive desire to materially accumulate—and lastly the practice of *pratikramana* (ethical introspection), which calls upon Jains to actively reflect upon their own daily minor transgressions (especially concerning *ahimsa*) and then aim to do even better.

However, though each of these practices contain rich ecological significance, it is important also to remember that Jainism originated first and foremost as a soteriological religious pathway, focusing on spiritual liberation at the personal level. Hence, as a number of recent scholars have been keen to point out, constructions of Jainism as being inherently 'green' or 'environmentalist' must be called into question,[3] for on the one hand issues of ecology have never been Jainism's primary or defining emphasis (or end-goal), and on the other, as John Cort has noted, Jain ecological practice is in a sense fundamentally estranged from modern environmentalism, seeing how the former is based on ancient teachings and the latter represents only a very recent "field of inquiry" (episteme).[4] Nevertheless, even if a fully-fledged modern Jain environmental ethic has yet to be formulated (or even if this is an impossible task in the first place), by no means should this prevent, at least in the opinion of this author, the looking upon of certain Jain teachings or practices as being, relative to our modern times, ecologically meaningful, applicable, or even perhaps quite timely and necessary. In fact, I would suggest that this latter potential represents a rather reasonable and vital potential, taking up as it does one of the key directives of UNESCO's 2000 *Earth Charter* declaration, which calls for modern individuals to "recognize and preserve the traditional knowledge and spiritual wisdom in all cultures that contribute to environmental protection and human well-being."[5]

Thus, in accordance with this directive, while first offering a brief inventory of Jain ecological thought and practice in general, this article will then analyze the eco-conscious dimensions of the modern Jain-inspired Anuvrat Movement—a social reform campaign established in 1949 by the late Jain (Shvetambara Terapanth) mendicant leader, Acharya Sri Tulsi. A widely celebrated and inspirational figure not only for modern Jains but also Indian citizens more broadly, a key teaching of Tulsi's was his view that positive social change depended first and foremost upon the ethical and spiritual conduct of *the individual*. Putting this philosophy into practice, Tulsi thus formulated for Anuvrat a series of 'small vows' (*anuvrat*) of ideal conduct, something he envisioned would beneficially impact both individual as well as society alike. Notably, among these 'small vows' was one fully devoted to ecological concern: namely, Anuvrat's final vow, vow eleven, which calls for practitioners to "refrain from such acts as are likely to cause pollution and harm the environment," and to avoid the "cutting down of trees" and the "wasting of water."[6] Beyond the explicit ecological significance contained within vow eleven, moreover, eco-conscious relevance was also expressed through several of Anuvrat's other vows as well, albeit on a more implicit level. Thus, exploring each of these vows in further depth as well as touching upon some of the basic history and philosophy behind Anuvrat, this article will also analyze Anuvrat's potential for ensuring ethical (and eco-conscious) behavior via its hallmark mechanism of vow restriction—a modality of arguably potent strategic and motivational value. Altogether, while first taking a brief inventory of Jain ecological practice in general, the article will then explore how when it comes to the modern eco-conscious imperative to "live simply so that others may simply live" (as the popular adage has it), there is indeed much that Anuvrat has to offer.

---

[3]　As one notable example, see (Cort 2002), "Green Jainism? Notes and Queries toward a Possible Jain Environmental Ethic."
[4]　(Cort 2002, p. 66).
[5]　(Pojman and Pojman 2008, p. 664).
[6]　Anuvrat code citation: accessible via http://www.anuvibha.in/Code-of-Conduct.htm.

## 1. Jainism and Ecology: A Brief Overview of Recent Scholarship

Prior to launching into our analysis of the Anuvrat movement, let us first touch upon the topic of Jainism and ecology in general. An important point to initially consider here is that, as with certain other Indic religious pathways, Jainism's *primarily ascetic* origins and characteristics have elicited a healthy degree of contemporary skepticism towards the basic ecological soundness of its teachings. Historically speaking, given the by and large dismal ecological track record of dualism and transcendence focused worldviews, both religious and secular, many modern folks have found themselves sharply in criticism of such world-devaluing perspectives, choosing (rather understandably) to favor more world-affirming and relational approaches instead. However, in line with Larry Rasmussen's notion of an "earth-honoring asceticism,"[7] considering the earth's current population and also the unsustainable rate at which humans currently participate in eco-destructive forms of material consumption, it is quite arguable that Jainism nevertheless offers some truly crucial ecological insight and value. Also, as another important preliminary point to consider, it bears noting that responding effectively to ecological crisis will require of humanity to make key changes at not just one but a variety of different levels: that is to say, *structurally* (making adjustments politically and economically, for instance); at the level of our various cultural and individual *thought-paradigms* (challenging anthropocentric, patriarchal, and racially prejudiced thinking, for example); as well as, on a more basic and external level, making the necessary changes (or sacrifices) to our own *personal lifestyle patterns*. And among these three distinct and arguably indispensable levels, it is especially this latter one—that of making key adjustments to our own (often heretofore eco-destructive) lifestyles—where I believe Jain teachings and practices may be especially valuable for the unique inspiration and insight they can offer.

Before launching into an analysis of specific teachings, however, it first merits discussing a basic divide which has recently surfaced within Jainism and Ecology scholarship. This refers to how ecological principles have recently been assessed vis-à-vis their modern relevance and viability, revealing in this respect a fundamental difference in approach. Though this basic methodological divide cannot be reduced to any one singular factor, I would suggest that it has much to do with how scholars tend to respond—whether more affirmatively or skeptically—to the following key series of questions (as posed by Christopher Chapple in his introduction to *Jainism and Ecology: Nonviolence in the Web of Life*):

> "Is this worldview compatible with contemporary ecological theory? How might a Jain ethical system respond to the challenges of making decisions regarding such issues as the development of dams, the proliferation of automobiles, overcrowding due to overpopulation, and the protection of individual animal species? Can there be a Jain environmental activism that stems from a traditional concern for self-purification that simultaneously responds to the contemporary dilemma of ecosystem degradation?"[8]

Though these questions are individually distinct from one another (and therefore in each case invite a distinct set of reflections to be made), it is worth noting that as a whole they have tended to be met by scholars of Jainism with either a prevailing sense of optimism, or otherwise a prevailing sense of hesitation. While on the one hand there exists a number of scholars (and scholar-practitioners) whose work presumes there being a rather workable compatibility between these two contexts, with whatever incongruities there may be being seen as either of a non-essential nature, or otherwise being eligible to be adaptively worked around; on the other hand such incompatibilities have tended to inspire a far more dubious (or at least cautious) sentiment. And because there are important distinct reasons upon which the latter feeling has been grounded, these different reasons should be briefly acknowledged. At the risk of oversimplification, I would suggest that they revolve around three primary concerns, in

---

[7]  (Rasmussen 2013, p. 252).
[8]  (Chapple 2002b, p. xxxvi).

particular—namely: (1) contextual incompatibility; (2) selective distortion; and (3) world-denial. Let us begin with the former:

(1) *Contextual incompatibility*: as already hinted at above, here we see a reluctance to assert that Jain religious teachings, formulated as they were during an entirely different era, are somehow nevertheless compatible with the various circumstances and concerns of modern environmentalism. As Cort has, for instance, noted:

> "The reason I say that there is no Jain environmental ethic is that environmentalism is a relatively new episteme worldwide. It has arisen out of a set of physical, technological, and increasingly moral and intellectual challenges of the past several centuries, but has attained its position as a distinct field of inquiry—an episteme—only within the past decades."[9]

The challenge that Cort poses here can be seen as a corrective reaction against the tendency for Jain religious teachings to be considered inherently and unquestionably "green" or "environmental," and that is to say, without such teachings first being shown to be able to adequately account for the various complex nuances and realities of contemporary ecocrisis (as involving, for instance, a variety of different structural, globalized, and patently modern phenomena). Hence, until Jain teachings are creatively reformulated or adapted in order to meaningfully speak to these more modern and complex realities and concerns, Jain ecological teachings are bound to remain relevant in only a somewhat limited or partially applicable manner. In short, for a Jain ecological ethic to be considered comprehensively sufficient to our contemporary circumstances, further creative envisioning and articulation would be needed.

(2) *Selective distortion*: here we see suspicions raised relative to how common portrayals of Jain ecological practice have heretofore tended to be constructed in an oversimplified and conveniently selective manner—for instance, all too often 'cherry-picking' out the exclusively positive dimensions (certain isolated practices or verses of scripture, for example), while simultaneously ignoring or downplaying some of the more problematic ones. The upshot here is that if Jain teachings are going to be assessed—and, by extension, more often than not celebrated—for the various measures of ecological value they contain, then also included within such an evaluative process should be a fair and even-handed accounting for whatever so-called 'shadow aspects' may be lurking. This general issue is also, it is important to note, not just a problem of selective distortion, but also *selective cooptation.* As Paul Dundas briefly explains, "Contemporary environmentalism seems to be a particular issue into which Indian religious traditions are coopted somewhat uneasily if their own highly ambivalent presuppositions about nature and the world are not fully taken into account."[10] Hence, as with the Jain perspective of *anekantavada* (non-absolutism; many-sidedness), Jainism and ecology scholarship needs to ensure that it takes into account more than just the rosier elements of tradition. Lastly, between warnings against cooptation and criticisms of cherry-picking, one may wonder if all of this simply boils down to scholars needing to do a better job to present a more fair and balanced portrayal of Jain ecological teachings—again, a sense of being willing to include the bad along with the good, and the questionable along with the clearly laudable—or if instead the entire endeavor might be considered to be more or less shipwrecked from the outset (that is to say, that the positive dimensions are not only *diluted* by the more problematic elements, but are instead either *undermined*, *canceled out, or are rendered no longer credible* by them). Though my own opinion on this leans more towards the former being true, I ultimately leave this as an open question for the reader.

(3) *World-denial*: this final point of skepticism involves Jainism's fundamental tendency for world-denial, a characteristic it shares with a variety of other Indic-samsaric religious pathways. Here, similar to above, there is a sense that whatever positive ecological aspects are identifiable within Jainism, in the larger scheme of things they are critically undermined by the Jain tendency for espousing religious and philosophical views of the more 'world-denying' sort. As Jeffery Long summarizes:

---

9    (Cort 2002, pp. 65–66).
10    (Dundas 2002, p. 111).

"Other scholars of Jainism, on the other hand, have called this view into question, arguing on the basis of Jain textual traditions that Jainism has more typically expressed a world-denying ethos of extreme asceticism which, far from positively valuing the world—and by implication, the physical environment—sees it as an obstacle to overcome. A re-envisioning of Jainism as a 'green' tradition therefore involves inevitable distortion."[11]

However, with this being said, Long also proceeds to point out:

"Although, as some scholars have pointed out, the j*iva*—like the *purusha* of Samkhya philosophy—implies a dualism as radical as the Cartesian dualism that has facilitated the Western devaluation and exploitation of the natural world as mere material for consumption, Jain practice would seem to belie this. Although [Jain] philosophy may be world negating, its practice issues in a negation of this negation: a profound mindfulness of one's environmental impact in life."[12]

Thus, Jainism's standpoint of metaphysical dualism—between *jiva*, sentience, and *ajiva*, insentience, as well as between *jiva* (here as a sense of one's indwelling soul) and that of one's body and their accumulation of karma—has *in practice* tended actually to be expressed rather innocuously, by and large managing to steer clear of the negative ecological ramifications that have so disastrously arisen from the widespread embrace of other such world-negating metaphysical dualisms—such as that of the Enlightenment-based Cartesian formulation. Indeed, while traditional Jain teachings have tended to discourage world engagement (largely on account of the fear of increasing one's own samsaric entanglement), at the heart of Jain ethics there has at the same time always existed an undergirding emphasis placed upon compassion (*anukampa*), nonviolence (*ahimsa*), and the willingness to radically minimize the negative impact one has upon their surrounding environment. And while it is true that a Jain's prerogative for the avoidance of accruing negative personal karma plays an important justifying role (or central motivation) behind such nonviolent and ecological practice, I believe a charge of non-altruistic individualism would be misplaced—failing to take into account the primarily *religious* (i.e., based on the core worldview of samsara, karma, etc.) and not ethical basis for such an emphasis. Still, when evaluating the ecological value of Jain teachings, it is important to take its dualist presuppositions into account, for such a worldview can indeed often lead to subtle negative ecological implications being expressed in both thought as well as practice (as has in fact sometimes historically been the case—even within Jainism[13]).

Altogether, I would suggest that the question of taking a more affirming versus a more skeptical approach towards the modern relevance of Jain ecological teachings is not so much a matter of one side being "right" and the other "wrong," as much as whichever direction one tends to lean towards natural carrying with it its own intrinsic advantages as well as shortcomings. And the reason I personally favor the more affirmative approach is because, simply put, I do not think it is actually such a stretch for these ecofriendly Jain teachings and practices—whether they need to first be creatively adapted, or not—to be made relevant to our contemporary circumstances. Furthermore, seeing how the vast majority of Jainism's teachings of ecological pertinence operate on more of a personal level of practice (again, addressing the eco-conscious lifestyle aspect), I believe this fact amplifies their potential for being adopted, or at least learned from, in a more or less universal (or trans-historical) manner. Of course, these questions also do not exist in a vacuum, and now with contemporary ecological crisis presenting a valence of superseding ethical urgency, like many others I believe it would actually be unwise to *overthink* or become *overly careful* in terms of our collectively gathering ecological inspiration from around the

---

[11] (Long 2009, p. 181).
[12] (Ibid., p. 182).
[13] For instance, among other examples, some scholars have I believe rightly criticized certain Jain animal shelters (*pinjrapoles*) as expressing a commitment to life-extension over and above the offering of comfortable (and humane by Western standards) living conditions.

world. Finally, it is also important to note here that, semantically speaking, asking the question "Does Jainism have a modern environmental ethic?" (in other words, one that is fully-developed and adequate to contemporary circumstances) is a far and away different question from asking "Are there teachings within Jainism that in our present day contain relevant ecological value?" Point being, while it is fairly easy to respond affirmatively to this latter question, the former presents a much harder case, and we should therefore be careful not to conflate the two.

## 2. Jainism and Ecology: A Brief Overview of Key Principles

Transitioning now into a brief overview of Jain ecological thought, let us begin with the Jain principle of nonviolence (*ahimsa*). Jainism's overriding emphasis upon this virtue is no less than foundational, as can be witnessed in the encapsulated traditional saying '*ahimsa paramo dharma*'—that 'nonviolence is the highest law' (or 'the highest religion'). A driving principle behind innumerable Jain teachings and practices, ahimsa necessitates going to great lengths in order not to kill (or for that matter, harm) life forms of any kind—including even the tiniest of microorganisms (*nigodas*). In this respect Jainism is known for the uncommon extremes in which its adherents strive to steadfastly fulfill this practice on a daily basis. Though certainly applying at the less stringent lay (or householder) level of practice as well, Jainism's radical commitment to non-violence is most clearly expressed by its mendicant practitioners in particular, whose entire lives become scrupulously regulated. As Vallely notes:

> "The entire logic of the ascetic's daily routine is dictated by the ethic of ahimsa, or nonharm. It is in interactions with the nonhuman world that the ascetics are most highly attentive, observant, and mindful—that is, when they are most quintessentially ascetic. Interactions with 'nature'—with the air, water, soil, and vegetation—define both lay and ascetic Jains by determining the boundaries of their ethical being."[14]

A large extent of what Vallely is referring to here comes down to Jainism's behavioral rules for self-control (*gupti*) and carefulness (*samiti*) which mendicant practitioners must at all times strictly adhere to, and which together become expressed at levels of one's bodily movements, speech and even inner thoughts. Presenting a prime historical example of this supreme degree of nonviolent punctiliousness is the enlightened Jain tirthankara (teacher; 'ford-maker') Mahavira, whose exemplary model is vividly recorded in the Kalpa sutra:

> "Henceforth the Venerable Ascetic Mahavira was houseless, circumspect in his walking, circumspect in his speaking, circumspect in his begging, circumspect in his accepting (anything), in the carrying of his outfit and drinking vessel; circumspect in evacuating excrements, urine, saliva, mucus, and uncleanliness of the body; circumspect in his thoughts, circumspect in his words, circumspect in his acts [ ... ] His heart was pure like the water (of rivers or tanks) in autumn."[15]

More than simply an ideal case dating back to an ancient bygone era, the extremely attentive level of nonviolent behavior that Mahavira heroically embodies here—that is to say, being 'circumspect' in all things, so as not to needlessly harm life forms of any kind—also very much percolates into the level of lay practice. Although when compared to the mendicant context, lay Jainism happens to be far more moderate and relaxed, Jain laity must nevertheless observe a number of measures that outsiders to this tradition would surely interpret as being radically nonviolent. Most notably, such measures include (among other such practices): the adherence to a strictly nonviolent dietary code (one that is vegetarian and avoids various soil-grown and bulbous foods); avoiding the use of leather, silk, and pesticides; choosing not to eat before dawn or after sunset (lest one's cooking fires or process of eating

---

14  (Vallely 2002, p. 199).
15  (Jacobi 1884, p. 260).

become hazardous to the hordes of airborne insects that are believed to be especially active at this time); and choosing a regular vocation in the world that is maximally non-violent (for instance, as a banker, or as a merchant).

Though the practice of *ahimsa* is largely justified by Jains through the prism of their own distinctive version of karma theory, where violent thoughts and actions are understood to attract the most spiritually-destructive forms of karma and should therefore be avoided as much as conceivably possible, nonviolent practice is also grounded in basic Jain perspectives on cosmology and biology. In this latter case, every life form is seen as being inviolably sacred and worthy of reverence—a recognition which entails, at the very least, each and every life form's inherent right to be respected and not harmed or interfered with. Supporting this deeply conservationist attitude is the belief that every living organism enjoys at its innermost level the existence of a spiritually boundless and luminescent *jiva*—with 'soul' providing an approximate (though not entirely accurate or unproblematic) translation of this term. Rather notably, the *jiva* is as an entity that is seen as being laterally equal to all others, where each and every *jiva* is believed to inherently possess an unlimited amount of energy, vigor, knowledge and bliss.[16] However, the major caveat to this is that the particular manifestations of each creature's bodily faculties and accumulation of personal karma serve to thickly cloud over and obscure such indwelling sentience and spiritual radiance, thus in reality manifesting a wide variance at which each creature may actually *access* and *experience* such innate luminosity from within. Still, from a Jain cosmological and biological standpoint each and every creature—no matter how macrobiotic (humans and others five-sensed living beings, for instance) or microscopic (e.g., tiny insects and microorganisms)—enjoys a natural right to live, flourish and karmically and spiritually evolve in the world. Put differently, and to summon the words of the pioneer of deep ecology Arne Naess, all of life according to Jains is seen to be imbued with 'autotelic value'[17]—that is to say, all of life has a *telos*, or driving purpose behind it, and therefore also (again to quote Naess) an "*equal right to live and blossom.*"[18]

In addition to nonviolence, another core Jain virtue commonly recognized for its ecological significance is that of *aparigaha* (non-possession). Seeing how this practice intersects with many of the ecological dimensions found in Anuvrat, I will only briefly touch upon it here, saving further analysis for later in the article; however, one point to note for the time being is that *aparigraha* refers both to the physical limiting of one's possessions, as well as one's achieving, in relation to whatever sparse possessions one may still possess, a general state of spiritual detachment. Thus, when practicing *aparigraha* a Jain must seek to avoid, in the words of P.S. Jaini, "harboring such false notions as 'this is mine' or 'I made that' and imagining that one can hold on forever to what he [or she] now 'has.'"[19] The starkest image of *aparigraha* is represented by the Jain Digambara ('sky-clad') male mendicant practitioner, who is required to endure a permanently naked existed and whose only personal items consist of a water gourd and, for the gentle sweeping away of insects, a peacock-feathered feather duster. While the rationale here also involves the spiritual opportunity provided through having to endure a sustained form of austerity by way of one's body being tested through its exposure to the elements and to the fluctuation of hot and cold weather; just as important as this is the overcoming of the shame of being naked (*lajja*)—considered by Digambara Jains to be a subtle but also key hindrance to spiritual growth—as well as the relinquishing and freedom from all possessions except that which is the very utmost essential. Additionally of note here is that for Jains the virtues of *ahimsa* and *aparigraha*, while enjoying an independent status from one another, are also considered to be closely interlinked. This is because feelings of possessiveness, no matter how subtle or seemingly insignificant, are understood to inevitably cause a sense of greed and selfishness to arise within an individual—with the added

---

[16] One small exception to this is that there are certain *nigodas* (one-sensed microscopic beings) which are considered to be unable to gain liberation. See for instance: Paniker 2010, *Jainism: History, Society, Philosophy and Practice*, p. 53.
[17] (Naess 2008, p. 225).
[18] (Ibid., p. 216).
[19] (Jaini 1979, p. 177).

potential for subtly violent forms of thought and action to then arise (since having possessions tends to cause one's feeling a need to guard and defend such possessions from others). While this point will be revisited later, for now it bears noting that the central ecological significance of *aparigraha* rests in the fact that having fewer possessions can drastically lessen the ecologically-damaging material footprint one brings about in the world. In essence, it allows a person to free of the accumulation of manufactured goods—products which are ultimately rather resource extractive as well as waste and pollution causing.

Similar but also distinct from the virtue of *aparigraha* is that of equanimity (*samyaktva*). From a Jain point of view, equanimity can perhaps be best understood though the individual who has risen above one's egotism and who has conquered the four fierce "passions" (*kashayas*) of anger, greed, pride and deceit, in order to arrive at a peaceful and—at some level awakened—spiritual disposition in life. Being non-reliant upon material need, the practitioner who embodies *samyaktva* may then be far more successful in terms of resisting the need to materially accumulate things. And thus we see how *aparigraha* and *samyaktva* both in turn robustly support one's adopting a lifestyle of material simplicity and eco-conscious minimalism. This practitioner of *aparigraha* and *samyaktva* naturally remains non-exploitative of one's environment, as illustrated by a classic Jain metaphor from the Dashavaikalika sutra. Here, the practice of Jain mendicant alms-seeking (*gochari*) is insightfully compared to the natural eco-friendliness of the bumblebee in its instinctual mode of gathering pollen in a sensitive and eco-conscious manner:

> "As a bumble-bee sucks pollen from flowers just a little at a time and satisfies its need without harming the flowers in any way, so are these absolutely detached *shramans* (Jain ascetics). They seek and gather faultless food from numerous houses exactly as the bumble-bees gather pollen from flowers."[20] (Dashavaikalika sutra 1.2-3)

Serving as one of the four mula sutras (root scriptures) that Shvetambara Jain mendicants must accustom themselves upon becoming initiated (*diksha*), the Dashavaikalika sutra lays out ideal conduct for such mendicants, in this case outlining the required protocol for the going about and begging for one's daily portion of food (*gochari*)—which is to say in a sustainable and non-avaricious manner. Besides illustrating the cherished Jain principles of contentment and spiritual detachment (the Jain mendicant not taking one's source of food for granted or becoming attached to one's benefactors), this verse is also taken to clearly endorse a sensitivity towards resource conservation and the importance of not overly imposing one's needs or desires to the detriment of any natural habitat. However, in order to do this, a Jain mendicant must assuredly first exhibit discipline (*vinaya*) as well as self-control (*gupti*) and equanimity (*samyaktva*).

And though less acknowledged from an ecological standpoint, the Jain practices of regular ethical introspection (*pratikramana*) and repentance (or forgiveness-asking—*prayaschitta*), also arguably offers some rather compelling eco-conscious significance. The practice entails first recollecting one's ethical transgressions over a recent limited period of time (especially relative to whatever lapses in nonviolent carefulness one may have succumbed to), and then expressing sincere contrition for these lapses, asking for forgiveness and also embodying the necessary mental and spiritual resolve in order for one's behavior to become properly rectified in the future. As a mendicant, this ritual must be practiced twice daily, while for lay Jains it is optional but encouraged as a highly meritorious rite. Both lay and mendicant Jains must also perform *pratikramana* during the last day of the eight-day annual observance (or ten, in the case of the Digambara tradition) of the very important Jain tradition of Paryushan/Das Lakshana. At the end of this 8 (or 10 day) annual observance, Jains take great care to ritually atone and ask forgiveness for their transgressions, which includes thoroughly reflecting upon one's behaviors

---

[20]    Dashavaikalika sutra 1.2-3. As printed in (Up-Pravartak, Shri Amar Muni n.d., p. 9).

over the last year and recognizing one's faults therein. As Jaini notes about this ritual (in a passage revealing the rite's laudably animal-friendly basis):

> "The admission of sins, and accompanying pleas for forgiveness (ksama), are directed not only to a teacher but to all of one's family and friends, irrespective of age or sex. Letters are written to those relatives and acquaintances not in attendance, repeating the same acknowledgements of wrongdoing and solicitations of pardon. Finally, the participant in a samvatsari extends his own forgiveness to all beings and asks that they grant the same favor to him; this is done by repetition of a famous verse which points up the real spirit of pratikramana—the establishment of universal friendship and goodwill: 'khamemi savvajive save jiva khamantu me/metti me savva bhuesu veram majjha na kenavi//I ask pardon of all living creatures; may all of them pardon me. May I have a friendly relationship with all beings and unfriendly with none.'"[21]

Thus, not only does this practice require a high level of moral accountability and the humility to admit and feel remorse over one's accumulated instances of subtle transgression, but it also, in the asking "pardon of all living creatures" reinforces a highly progressive animal-friendly ethic, wherein not just humans but all creatures must be supplicated for forgiveness.

Ecologically speaking, I would suggest that the main significance of this rite is actually two-fold. On the one hand, the rite clearly instills an ethic of deep compassion and care towards one's fellow living beings in the world, providing the opportunity for Jains to really pause and consider how one might improve one's ethical consistency in this regard. Seeing how eco-conscious living requires a strong sense of living mindfully and intentionally in the world (as well as a commitment towards proceeding cautiously), I would suggest that such a practice as this offers some excellent potential for emulation at the level of eco-conscious practice (those who are pursuing an eco-conscious life reflecting back at regular intervals on how well one is managing to meet the ideal, and why again one is again engaging such a practice in the first place). Secondly, also of significance here is the relationally-oriented ethic that the rite invokes, wherein one actually takes some time to dwell upon the fundamental sanctity of all other life forms, thereby empowering one's future actions to be all the more naturally motivated. Highlighting the essential importance of such a relationally-based ethic, as ecology pioneer Aldo Leopold once noted: "We can be ethical only in relation to something we can see, feel, understand, love, or otherwise have faith in."[22] One is naturally here reminded of the importance of arousing *genuine feeling* towards nature and living beings, as opposed to acting out of a cold and heartless sense of mere obligatory action or perfunctory routine. Such a proactive and affirming attitude as this also happens to be a fundamental insight of spiritual ecology, and as Bhagchandra Jain 'Bhaskar' succinctly states to this effect (in his chapter on "Ecology and Spirituality in the Jain Tradition"), "The global ecological crisis cannot be solved until a spiritual relationship is established between humanity as a whole and its natural environment."[23] Thus, the Jain *pratikramana* and *prayaschitta* rites thereby provide a regular opportunity for cultivating this kind of interconnected, compassionate and attentive relationship with the non-human world, which again requires first opening up the necessary space in one's life order to allow such feelings to truly and deeply permeate. Hence, in committing themselves to the *pratrikramana* rite, Jains may regularly recognize how we as humans, as Chapple puts it, "have been given the special task and opportunity to cultivate increasingly rarified states of awareness and ethical behavior to acknowledge that we live in a universe suffused with living, breathing, conscious beings that warrant our recognition and respect."[24] And with this being said, and also with this larger

---

[21] (Jaini 1979, p. 216).
[22] (Leopold 2008, p. 168).
[23] (Bhaskar 2002, p. 170).
[24] (Chapple 2002a, p. 130).

brief survey of Jain ecological practice coming to a close, let us now shift our attention over to the Anuvrat Movement—which iitself offers a great deal of eco-conscious relevance and inspiration.

## 3. Anuvrat's Background, Philosophy and Ecological Import

Aiming at the "moral and spiritual regeneration" of Indian society,[25] the Anuvrat Movement was launched on 2 March 1949—a mere two years after India's gaining Independence as a sovereign nation-state. While Tulsi's impetus for the movement cannot be boiled down to any one singular factor, it is fair to say it revolved around what he perceived to be a general, multifaceted moral deterioration within Indian society. Prior to this, for the sake of achieving Independence a strong spirit of national unity had been forged across Indian society—a sense of self-sacrifice and the putting aside of differences in order to unite around a common, collective cause. However, not long afterwards this positive and unified spirit had largely dissipated, with outbreaks of selfishness and bitter divisiveness taking root instead. Though freed from the oppressive shackles of British colonial rule, Indian society had nevertheless become newly afflicted by what Kanakprabha describes as "fissiparous tendencies,"[26] a "degenerative miasma"[27] and a "scourge of all pervading corruption."[28] And as Tulsi himself summed up the situation: "Indiscipline, craving for high position, ambitiousness, regional and language controversies—these cropped up in the wake of freedom. All these were responsible for the growing deterioration in character and increasing mental agonies of the public."[29] Hence, seeking to counteract such selfishness, "communalism (i.e., political parochialism), and general moral deterioration, Tulsi conceived of Anuvrat as a way to address this issue at its very root: namely, what he saw as being a lack of self-restraint and ethicality being expressed at the individual level.

It was here that, in order to help remedy the situation in concrete and specific terms, Tulsi came to propose Anuvrat's list of individual moral and spiritual vows. Tracing the historical evolution of this list, originally thirteen in number, the vows soon became vastly expanded, ballooning all the way up to a compendious eighty-four, before finally becoming distilled back down to a more manageable set of eleven. Important to note here is that, as a Jain mendicant practitioner, Tulsi chose to model the vows largely along traditional lines, with Jainism's five "great vows" (incumbent upon mendicants) and its five "small vows" (incumbent upon layfolk) providing the primary ideological inspiration. Specifically, these five categories of vows include *ahimsa* (non-violence), *aparigraha* (non-possession), *asteya* (non-stealing), *brahmacharya* (celibacy), and *satya* (truthfulness)—these five being the same as the five *yama* (self-control) vows of Classical Hindu Yoga. However, in lieu of spiritual liberation (*moksha*, *kevala*) being the ultimate end goal (as is the case within Jainism more broadly), for Anuvrat it was instead moral regeneration and social uplift that provided the ultimate and prevailing *telos*. And while no doubt shaped by a distinctively Jain religious standpoint, the movement nevertheless opened itself to any and all who might be willing to pledge, irrespective of their particular gender, caste, religion, or ethnicity. This spirit of universalism and nonsectarian inclusivity also deeply suffused Tulsi's own personal attitude: "I am a human being first and then a religious man," he once explained, "as regards my being a Jain and Head of a Jain sect, I put these positions in the third and fourth places, respectively."[30] Thus, although historically speaking Jains themselves have accounted for the majority of Anuvrat's overall base of practitioners, the movement has also gained extensive support and participation from a large number of Hindus, Muslims, Sikhs and Christians, as well as those of a more non-religious persuasion.[31] Lastly, as far as Anuvrat's current representation is

---

25 (Gandhi 1987, p. 6).
26 Kanakprabha, Sadhvipramukha. 1985. *Acharya Tulsi: A Life Sketch*. Translated by Narendra Moray.
27 Ibid., p. 5.
28 Ibid., p. 6.
29 (Mahaprajna 1994, pp. 3–4).
30 (Mahapragya 2017, p. 14).
31 (Mahaprajna 1994, p. 17).

concerned, the movement is currently being promoted by four main entities, in particular: first and foremost by Acharya Sri Mahashraman (the current head of Terapanth and official present-day leader of the movement)[32]; by Terapanth's order of semi-renunciant global missionaries (called *samanas* and *samanis*, respectively); by the Anuvrat Mahasamiti[33]; and lastly by the Anuvrat Global Organization (ANUVIBHA)[34]—a volunteer-led organization that promotes Anuvrat on an international stage and that participates in the U.N. as well as other such key global assemblies.

With Anuvrat's main hallmark and emphasis consisting of its personal code of conduct—the eleven vows, that is to say—the overarching guidelines for the movement are as follows:

Aims

1. To inspire people to observe self-restraint irrespective of their caste, color, creed, country or language.

2. To establish the values of friendship, unity, peace and morality.

3. To create a society free from all kinds of exploitation.

Means

1. To inspire a maximum number of people to be anuvratis (those who pledge themselves to observe basic vows in their daily life).

2. To bring about a revolution in thinking and action.

Eligibility

All those who believe in leading a pure life will be entitled to become anuvratis.

Code of Conduct

1. I will not kill any innocent creature.
I will not commit suicide.
I will not commit feticide.

2. I will not attack anybody.
I will not support aggression.
I will endeavor to bring about world peace and disarmament.

3. I will not take part in violent agitations or in any destructive activities.

4. I will believe in human unity.
 I will not discriminate on the basis of caste, color, creed etc., nor will I treat anyone as an untouchable.

5. I will practice religious tolerance.
I will not rouse sectarian frenzy.

6. I will observe rectitude in my dealings with other people.
I will not harm others in order to serve any ends.
I will not practice deceit.

7. I will set limits to the practice of continence and acquisition.

8. I will not resort to unethical practices in elections.

9. I will not encourage socially evil customs.

---

[32] For a general profile of Mahashraman available on the web, see: www.acharyamahashraman.in/profile/anurvat-anushasta.
[33] To visit this organization's official website (available in Hindi), go to: www.anuvratmahasamiti.com.
[34] See official ANUVIBHA website at: www.anuvibha.in.

10. I will lead a life free from addictions.
I will not use intoxicants like alcohol, hemp, heroin, tobacco, etc.

11. I will do my best to refrain from such acts as are likely to cause pollution and harm the environment.
I will not cut down trees.
I will not waste water.

Anuvrat Sadhana (regimen of proactive practices)

1. I will practice Preksha Meditation.

2. I will have a reconciliatory attitude for the sake of a peaceful domestic life.

3. I will practice restraint in individual possession and consumption.

4. I will exercise control over eating.

5. I will practice diligence, self-reliance and simplicity.[35]

Among the eleven vows of Aunvrat's code of conduct, clearly it is the eleventh one which is the most directly and explicitly ecological. Though at the time when Anuvrat was established, India (and the world at large, for that matter) was experiencing far different ecological circumstances than now, not only do the four basic injunctions of vow eleven (to not pollute, harm the environment, cut down trees, or waste water) remain fully relevant to today's world; if anything their relevance over time has only greatly intensified. For example, relative to our present times, the eleventh vow's requirement not to cause pollution or harm the environment naturally intersects with a modern person's duty for strictly limiting (if not outright avoiding) behaviors identified as being the most responsible for worldwide climate change. In light of this, the fulfillment of this eleventh vow would therefore today entail a wide range of eco-conscious behaviors, include (to name just a few): the curbing of one's accumulation of personal goods, the boycotting of eco-unfriendly products and companies, and the reduction of carbon dioxide emissions through one's regularly choosing, for instance, public over private modes of transportation. With regard to conserving water, moreover, it would also entail such considerations as, "What kind of toilets and showerheads do people buy? [And] how often do they wash their cars or water their lawns?"[36] as Shivani Bothra has noted (in her own work on the Anuvrat Movement[37]). Similarly, when regularly and meticulously practiced, seeking to limit one's use of water while washing the dishes or brushing one's teeth represent two other impactful water-conserving behaviors to consider—and these, too, would fall under the jurisdiction of vow eleven.

Still, however much Anuvrat's vow eleven can be said to encompass the fuller range of ecological concerns and practices, by no means is it the only one that encourages (at some level of another) an eco-conscious mode of life. In looking to vows seven, ten and four, for instance, there is plenty of eco-conscious potential that presents itself here as well, albeit perhaps in a more implicit manner. Let us now turn our attention to these additional dimensions of eco-conscious relevance.

Beginning with vow seven, to "set limits to the practice of continence and acquisition,"[38] here we must consider the more ecologically pertinent 'acquisition' aspect of this vow, with this dimension being intimately related with the cornerstone Jain virtue of *aparigraha* (non-possessiveness)—one of Jainism's five main mendicant and lay vows, as touched upon earlier. Seeing how the emergence of a variety of dimensions of worldwide ecological crisis have been shown to critically hinge upon mass trends of material acquisition and human consumption, the ecological relevance of vow seven is actually quite striking. For instance, just to name a few of the most serious of the ecological ramifications that this

---

[35] Anuvrat code citation: accessible via http://www.anuvibha.in/Code-of-Conduct.htm.
[36] (Bothra 2013, p. 58).
[37] For an excellent work exploring the Anuvrat Movement from a wider investigative lens, see (Bothra 2013). This Master's thesis is currently available as an open-access online resource.
[38] Anuvrat code citation: accessible via http://www.anuvibha.in/Code-of-Conduct.htm.

human desire to acquire and consume has largely been complicit in producing, these include (among others): global deforestation, rampant pollution, and, across the globe, the build-up of unprecedented amounts of ecologically-harmful waste material (the Great Pacific Garbage Patch representing but one conspicuous example). Beyond producing these disastrous effects alone, however, such widespread consumptive, consumeristic behaviors also inevitably connect back to the issue of anthropogenic climate change. To speak to this latter point, as one leading environmental research organization has recently found,[39] in the U.S. "an estimated 42% of [the nation's] total greenhouse gas emissions come just from the provision of consumer goods [alone]."[40] To contextualize this staggering statistic, beyond the more obvious concern of how such a massive stockpile of goods can possibly be responsibly disposed in the first place, what most consumers fail to take into account here is how mass patterns of consumerism also inevitably create effects that are said to be "up-stream" from one's actually purchasing and using a certain product—factors which include, for instance, the highly energy-intensive processes of materials manufacturing (and to a lesser degree, packaging), as well as the severe ecological repercussions that come with shipping goods across sometimes extraordinarily vast distances. And though these lesser-acknowledged effects may receive less critical attention (being more or less chalked up to the perceived-as-necessary realities of global and regional commerce), as this NWEI statistic shows, in terms of their producing vast quantities of carbon dioxide they are essentially just as culpable a factor as any other. Clearly, then, to live eco-consciously in today's world must invariably include a sense of "setting limits" to one's practices of acquisition, which is precisely what Anuvrat's vow seven calls for. Whether 'up-stream' or 'down-stream,' in other words, obviously no ecologically-damaging effects can result from goods which happen not to be purchased in the first place.

As for vow number ten, to "lead a life free of addictions,"[41] though for Tulsi this was meant to specifically refer to the use of intoxicants such as "alcohol, hemp, heroin, tobacco, etc.," ecologically speaking it can also be arguably extended to encompass our various *societal addictions* as well. Naturally, this would then implicate developed (and developing) nations from around the world for their more often than not chronic addiction to fossil fuels—most notably, petroleum, coal, and natural gas. Here, the applied basis of vow ten again involves, as was seen also with vow seven, a general calling for self-restraint and the setting of limits to an individual's—or in this a society or specific nation's state's—habitual patterns of consumption.

Lastly, vow four's mandate to "affirm human unity" reinforces the essential importance of striving for global solidarity as we collectively aspire towards creating ecologically sustainable societies, and that the choosing of self-interest of divisiveness (over that of self-sacrifice and cooperation) may ultimately, if unchecked, become the central cause of our own demise. On the other hand, vow four also serves as a reminder that environmental destruction and the effects of climate change tend to affect people across the world disproportionally, with citizens from so-called "third world" and equatorial nations, for reasons both political and climatological, all too often bearing the immediacy and brunt of the negative effects. Most notably, such disastrous effects include more intense and more frequent hurricanes and famines, vast desertification, the rising of sea levels, unprecedented levels of flooding, and a variety of other such life-endangering phenomena. In this respect, Anuvrat's call for human unity thus requires the broadening of one's ecological conscience and purview to extend beyond one's immediate region or nation-state alone, thereby encouraging a far more inclusive and just vision of our world as an interconnected and interdependent whole.

As a final point of analysis, although not directly related to ecology, Anuvrat's prohibition against feticide (as contained in the first vow) needs to be briefly acknowledged here, as I would imagine many Western reproductive-rights advocates may naturally perceive this aspect to be a surprising,

---

[39] Namely, the state of Oregon's Department of Environmental Quality (DEQ).
[40] (Mihm 2017, p. 48).
[41] Anuvrat code citation: accessible via http://www.anuvibha.in/Code-of-Conduct.htm.

unnecessarily polarizing, or perhaps even reprehensible component of Anuvrat's overall ethical vision. Putting aside judgments towards its moral justifiability one way or another, however, it is important to realize that this injunction against feticide can only very loosely be interpreted as a weighing in upon the pro-life versus pro-choice debate that continues to today feverishly divide Western (and especially American) political landscapes. That is to say, far more than advocating for the right of embryonic human life over that of women's bodily and reproductive rights, the vow is more simply a logical extension of the all-important Jain imperative for "*ahimsa paramo dharma*"—that nonviolence (in relation to all life-forms) is one's supreme ethical and religious duty. Furthermore, one must also note that for Tulsi the vow against feticide had the additional significance of aiming to lessen India's alarmingly high rate of female infanticide, which is an issue that remains of major moral concern even today.[42] But the more essential point to keep in mind here is that for Jains the non-killing and non-harming of all life forms represents a moral imperative that must be taken extremely seriously—whether pertaining to human embryos, animals, or for that matter even insects and microorganisms. Thus, when evaluating this vow from an outside cultural lens, one must therefore remember that Jains tend to be overridingly "pro-life" at a fundamental level, but for reasons typically very different than, say, what the general outlook of a pro-life American Christian would be.

## 4. Anuvrat: Ethical Vow-taking and Eco-conscious Living

In her classic work *The Body of God: An Ecological Theology*, eco-feminist scholar Sallie McFague points out just how intrinsically complex the issue of global ecological deterioration happens to be, considering the radical extent to which we humans are ourselves at the very heart of the problem: "Ecological deterioration is subtle and gradual: it involves the daily, seemingly innocuous, activities of every person on the planet ... We are, then, dealing with a wily, crafty enemy: *ourselves*, as the perpetrators of ecological crisis."[43] As unfortunate and inconvenient as it may be, there is much that can pointed to in support of McFague's point here. For one, we must come to terms with the stark truth that over the last half-century (between 1970 and 2016) our global population has nearly doubled in size, thus in the process putting unimaginable strain upon the earth's overall carrying capacity, its ability to withstand humanity's effects amid such unprecedented population growth. If this were not enough, however, according to the Northwest Earth Institute (NWEI),[44] during this same period of time the extraction of the earth's natural resources not only doubled (as one might reasonably expect), but has in fact more than tripled.[45] Thus, with McFague's so-called "seemingly innocuous" human activities only set to further compound and multiply as time goes on, the need for a greater embrace of eco-conscious living becomes only all the more necessary and urgent. As religion and ecology scholar Larry Rasmussen puts the matter, "in a humanly-overpopulated world, asceticism is a mandatory pathway to sustainability ... we must transform ourselves from nature's children to nature's guardians by learning to say 'enough' to ourselves."[46] Adding to this argument, Alex Mihm from the NWEI advocates for essentially the same approach, though in has case framing the matter in terms of 'simplicity,' (that is to say, jargon devoid of religious overtones): "Simplicity, as the crux of thoughtful consumption," Mihm states, "is not hollow, feel-good fluff devoid of scientific substance. According to the latest research on materials management, it is our best bet for how to live in accord with the finite limits of our only world. It is our past, and it will need to be our future."[47] Thus, though approaching the issue of ecological crisis from rather distinct religious and philosophical standpoints, it is important to note how McFague, Rasmussen and Mihm each happen to agree about eco-remedial

---

42 (Bothra 2013, p. 43).
43 (McFague 1993, p. 3).
44 A non-profit think tank based out of Portland, Oregon that progressively promotes grassroots ecological efforts.
45 (Cagle 2017b, p. 15).
46 (Rasmussen 2013, p. 252).
47 (Mihm 2017, p. 48).

action needing to entail, at a very basic level, the curbing of normal, heretofore taken-for-granted human behaviors. As a Christian eco-feminist (McFague), a proponent of "earth-honoring asceticisms" (Rasmussen), and someone who could be said to broadly representative of the standpoint of secular humanism (Mihm), what each of these three end up confirming in their own way is once again the need for modern individuals to "live simply so that others may simply live."

By no great coincidence, the prescriptive consensus stated above happens also to closely parallel what Tulsi had positively envisaged as a "restraint-oriented society"[48] taking shape, by which he meant one predominantly comprised of individuals who have spiritual and ethical willpower, are able to practice self-control, and are committed to living in accordance with high moral standards. Relative to living eco-consciously, moreover, such "restraint-oriented" individuals are able to more or less keep in check their own selfish and greedy egoic passions (known in Jainism as the four *kashayas*) and additionally be able to model the Jain virtue of *aparigraha* (non-possessiveness),[49]–with both of these values also being amply incorporated into Tulsi's vision for Anuvrat. However, not only does Anuvrat itself clearly discourage a greedy, possessions-laden eco-unfriendly lifestyle, but rather notably it also, through the key mechanism of vow-taking, provides the essential means by which self-restraint and simplicity *may actually be successfully implemented in practice.* This point cannot be emphasized enough. For while it is one thing to be able to simply recognize the vital importance of eco-conscious living, it is quite another thing altogether to be able to actually embody the necessary mindset and life-habits to follow through and do it. And in this respect, ethical vow-taking (as modeled by Anuvrat) can provide some truly vital assistance.

This latter point becomes arguably all the more significant when we consider basic human psychology, and just how complacent and backsliding humans tend to be when it comes to our various efforts towards self-growth and ethicality—whether pertaining to eco-conscious living, or otherwise. For example, observing how "every man likes to swim along the current,"[50] Tulsi also once noted how "many people want to imbibe morality but are forced to give up half way in the face of adverse circumstances."[51] This is a key point, for as we all know adversity is woven into the fabric of life, and nowhere did Tulsi ever say that becoming an ethically scrupulously citizen would be easy, just as the same is true for one's achieving a high standard of eco-conscious living. As eco-psychologists Allen Kanner and Mary Gomes point out, on a similar note: "There is a great deal of loss involved in giving up the fantasy of a consumer paradise or in falling out of love with technology. Alternative, more sustainable ways of living are bound to appear boring and perhaps even depressing in comparison. Doubt and despair will emerge as people ponder whether change is possible or worth the effort."[52] Hence, as Kanner and Gomes here recognize, for the vast majority of us breaking out of our normal, 'business as usual' eco-destructive life patterns may be a far more difficult endeavor than we might at first imagine it to be. Just consider, for instance, all that the average modern person is ultimately up against: one's own selfish egotism, the indwelling forces of human desire, apathy and laziness, the tendency for becoming distracted at a moment's notice—and now in our modern times techno-addicted besides.

However, arguably presenting just the right tool for overcoming such forces of adversity and inertia, what ethical vow-taking enables a person to do is essentially "draw a line in the sand" relative to his or her own ongoing personal behavioral patterns—something which amounts to a significant (if not decisive) overall strategic advantage. Tulsi thus recognized how, in a clearly demarcated way (i.e., one either honors each of the Anuvrat vows, or one does not), ethical vow-taking has the power to

---

[48]    (Bhatnagar, R. P., ed. and Transl 1993, p. 7).

[49]    From a Jain perspective, the practice of limited possession (*aparigraha*) is of vital ethical and spiritual import; just as it frees a practitioner from the concomitant worry and attachment that comes with owning and looking after numerous personal possessions, it is also understood to prevent the egoic, *himsic* (violent, aggressive) tendencies that surface as one seeks to safeguard and protect one's possessions from others.

[50]    (Tulsi 1998, p. 16).

[51]    Ibid., p. 16.

[52]    (Kanner and Gomes 2005, pp. 89–90).

ensure that one becomes, in essence, fully ethical (or eco-conscious) in all that he or she does—thereby becoming a sealed container, as it were, for the expression of ethical (or eco-conscious) behavior alone. The efficacy of this strategy has much owing to the power of one's basic sense of vow commitment and integrity of word: through the psychologically motivating mechanism of vow-taking, one may helpfully latch onto something tangible and thereby enter into a binding covenant one is not likely to casually defy. The potency of this insight—that is to say, the crucial role that vows can play towards ensuring either an ethical or eco-conscious lifestyle—has also recently been affirmed by the Anuvrat Global Organization (ANUVIBHA) as part of its participation within the UN Sustainable Development Platform:

> In order to ensure the eco-sustainability of the future and equitable distribution of the natural resources among all sections of people on the earth, the ANUVRAT Code of Conduct has been laid down by the founder of the Movement . . . Our organization believes that every human being irrespective of his caste, creed, nationality instinctively fulfils his pledges and keeps vows. Once a person takes a vow voluntarily he keeps it.[53]

Also significant to note here, moreover, is how such a strategy of behavioral containment has a meaningful basis within the word '*anuvrat*' itself. The term '*anu*' means 'small' while '*vrata*' means 'vow,' with the latter term being derived from the multivalent Sanskritic root '*vri*'. Among the vast nexus of meanings that exist for '*vri*,' some primary connotations include: to 'check', 'ward off', 'surround', 'obstruct,'[54] and, according to Jain scholar Padmanabh Jaini, 'to fence in' as well.[55] By what I would suggest is no great coincidence, these etymological connotations are quite helpful and clarifying towards elucidating the core strategy of the ethical-ascetic vow, wherein the transformation of either self or society can be radically accomplished through the 'checking' or 'fencing in' of a specific set of behavioral parameters. Concerning the philosophy of Anuvrat, for instance, Tulsi happened to express such a rationale of 'checking and 'fencing in' in the following manner: "It is through this self-imposed form of restraint that true resistance to temptation develops and self-control grows. Activity becomes pure to the extent to which impure elements are thus kept out. The main aim of the Anuvrat movement is thus to help develop in each individual the power of self-protection against the infection of impure conduct."[56] Thus, through either fully abstaining from (or otherwise watchfully moderating) such 'impure' thought-habits or behaviors—'impure' here simply referring to thoughts and behaviors deemed to be either unethical or spiritually damaging—these negative tendencies may in essence be effectively 'warded off,' while ethically or spiritually "pure" thoughts and behaviors are, in the meantime, simultaneously preserved (or 'fenced in'). As Jainism scholar Christopher Chapple notes about this kind of ethical-ascetic potential (in his 2008 article on "Asceticism and the Environment"):

> Ethical behavior serves as a corrective to address past wrongs and as a way of forging new pathways in the present to guarantee future states of auspiciousness. By sloughing off old impure behaviors and taking on new pure activities, both the individual and the society benefit. By skillfully applying precepts such as nonviolence and minimizing one's possessions in the context of one's ecological footprint, asceticism helps improve not only oneself but also the world.[57]

Thus, getting back to Rasmussen's notion of "earth-honoring asceticisms," as Chapple here notes, there is arguably a great deal that ascetic-ethical modalities can offer the world; and, as with Anuvrat, this is true on both the personal as well as the collective levels of existence. And though far from

---

[53] Anuvrat Global Organization (Anuvibha). *UN Sustainable Development Goals Knowledge Platform*. https://sustainabledevelopment.un.org/index.php?page=view&type=20036&menu=1561&nr=55363 (accessed on 15 March 2019).
[54] (Williams, Monier n.d.).
[55] (Jaini 1979, p. 169).
[56] (Gandhi 1987, p. 4).
[57] (Chapple 2008, p. 524).

being the only system—in India, or elsewhere—to utilize such a strategy, Tulsi's vision for Anuvrat nevertheless provides a compelling blueprint for such a transformative method of practice.[58]

Although concrete examples that illustrate the value of a vow-based strategy of eco-conscious 'fencing in' could be listed out almost endlessly, a single potent example comes from Portland, Oregon, in the form of a married couple who are noteworthy for their radically committing to an impressively eco-conscious lifestyle. Setting a powerful example for us all, what this particular couple—for the sake of anonymity, let us refer to them as the Taylor's—elected to do was to drastically limit their own production of personal waste, managing to restrict an entire year's worth to fit within a single 35-gallon standard garbage can.[59] Now, perhaps this achievement may not sound all that heroic or game-changing at first, but if we are to assume that the average American couple fills the equivalent of one such garbage can each and every week (which in the U.S., at least, is not at all unreasonable to assume), this would mean that the Taylor's have, in effect, essentially reduced their annual trash output by a rather staggering factor of 1/50th the standard amount. In a very concrete manner, they elected to 'draw a line in the sand' relative to their own ethical and eco-conscious habits and behaviors, and in this endeavor they ended up being wildly successful. Thus, while not self-professing *anuvratis* (followers of Anuvrat) themselves, the Taylor's admirable level of eco-conscious commitment nevertheless vividly illustrates the highly effective and transformative potency of the '*vrata*' principle—as one works to, along the same lines of the Anuvrat strategy, 'fence in,' 'ward off,' and 'draw a line in the sand' relative to one's own core ethical habits and behaviors.

## 5. Conclusions

In sum, as a potent vision for spiritual and ethical vow-taking, Anuvrat has much to offer the urgent contemporary task for eco-conscious living. Within Anuvrat's overall code of conduct, this aim is most directly and explicitly spelled out via its eleventh vow in particular, although with this being said several of Anuvrat's other vows contain some significant ecological potential as well. Upon reading this article one point that should stand out for the reader is how Anuvrat's strong eco-conscious potential can be largely credited to its incorporation of the potent modality of ethical vow-taking itself—a strategy that offers potentially decisive advantages in terms of overcoming natural psychological obstacles, life's adversities, and various engrained forms of resistance in the world. Finally, as a case study of Jain ecological practice, Anuvrat presents a compelling example of what Jain religious teachings can beneficially offer to broader contexts of ecological discourse and practice—and indeed at a time when such outside insight and inspiration is urgently needed. With this being said, I close this article with a quote from Tulsi which I believe greatly reverberates relative to our current ecological circumstances in the world: "there is after all a limit to everything. This situation has now touched the extremity. It should be propitious to be alert now. The age is awaiting [those] of courage capable of changing the tide once and for all."[60]

**Funding:** This research received no external funding.

**Acknowledgments:** I would like to sincerely thank Lexington books for granting permission to reproduce Sections 3 and 4 of this article from a forthcoming work of mine: namely, the book chapter "Acharya Tulsi, Anuvrat, and Eco-conscious Living," as featured in *Beacons of Dharma: Spiritual Exemplars for the Modern Age*. (Edited by Christopher Patrick Miller, Michael Reading and Jeffery D. Long. Lanham, MD: Lexington, 2019). Also, for his invitation to contribute to this special journal issue, I would like to express my deepest thanks to Dr. Leslie E. Sponsel. And finally, for providing insightful constructive feedback to an earlier version of this article, I would like to sincerely thank the two anonymous peer reviewers.

---

[58] Naturally, the utilization of vow-taking for societal transformation within India did not originate with Tulsi, a point that Tulsi is himself quick to openly acknowledge and celebrate: "The tradition of taking vows is an old and time-honoured one in this land. I do not therefore claim to have brought a new institution into being. All that I can claim is to have put new life into the moral heritage that we possess and to have adapted it to the needs of our times." (Gandhi 1987, p. 7).

[59] (Cagle 2017a, p. 51).

[60] (Tulsi 2013, p. 131).

**Conflicts of Interest:** The author declares no conflict of interest.

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
