# Peer review of "The Anuvrat Movement: A Case Study of Jain-inspired Ethical and Eco-conscious Living"

_religions, doi:10.3390/rel10110636_

Round 1

Reviewer 1 Report

7. Reword. Sentence is a run on. 

24. Please check the style guide. I think this supposed to be in text citation. Notes are for parenthetical notes to the readers. 

27. Use of KOSHER is inappropriate and too colloquial. 

31. See Note 24. And throughout paper. Use in text citation. Unless style guide says other wise. Also note should be numerical not Roman. 

62. I don't think oathsworn is a word. 

Introduction feels too long. Too focused on outlining each vow. Perhaps a shorter intro with overview of paper, and then a background section on the Founder, the movement, and then the details about vows? 

90 Don't recommend someone else's work on your subject. Reference the work, but stick with your own analysis and summary. 

Over all very well presented. Check formatting, some colloquialisms and rearrange intro. 

Author Response

Thank you very much for reading this article and for your helpful review.

I went through and made a number of significant changes to the article. And with respect to your specific comments, here are my responses: 

7) Sentence has now been shortened

24) In checking the style guide I now see the desired format (it appears in-text citations are indeed what is called for). I was unable to find a way to convert from superscript to in-text bracketing [], but will check with Ms. Kiki Zhang with respect to the step. Either in my next round of editing, or otherwise in the proofreading stage, I will be sure that this gets switched around.

27) Good point. Deleted and chose a different word.

31) I changed the roman numerals to numbers, but still need to switch to the in-text format

62) correct, my mistake. I deleted it. Also, introduction has now been clarified and shortened

90) Fixed

Reviewer 2 Report

This is a fine paper on the important topic of the Anuvrat movement and its ongoing relevance to the current ecological crisis.  It is well-researched, with most of the major literature on this topic having been included in the bibliography.

Should the author have time, it would be nice to engage with the articles by John Cort and Paul Dundas in Chris Chapple's edited volume Jainism and Ecology (a volume which this paper cites) which raise critical questions about discontinuities between Jain environmental activism and traditional Jain soteriology.  These questions are not insuperable obstacles, and suggestions for how to answer them have been made by Jeffery Long in the final chapter of Jainism: An Introduction.  While not essential, engagement with these questions would make the paper more comprehensive as a philosophical statement in favor of the Jain ecological ethic.

Author Response

Thank you very much for reading this article and for your helpful review. I also appreciated your compliments on it.

I went through and made a number of significant changes to the article. And with respect to your specific comments, here are my responses: 

I have now presented a fuller methodology section, which includes engaging the work of Cort and Dundas. Actually, I'd been meaning to integrate these two chapters into this article's 2nd draft anyway, so I appreciate the extra motivation for doing so. Admittedly, I struggled from the outset in terms of how to negotiate such discontinuities. I have my own position on these questions clearly in mind, but I found this to be a difficult matter to address and sensitively negotiate, especially with the limited space available. If you happen to have any further recommendations for deleting, reworking, or adding to any part of this new section, please let me know. I also integrated the work of Jeffery Long and found the recommended section of his work to be very pertinent and helpful.